# Single-cell analysis reveals host S phase drives large T antigen expression during BK polyomavirus infection

**Jason M. Needham¤, Sarah E. Perritt, Sunnie R. Thompson** \*

Department of Microbiology, University of Alabama at Birmingham, Birmingham, Alabama United States of America

¤ Current address: Department of Cell Biology and Molecular Genetics, University of Maryland, College Park, Maryland USA

\* sunnie@uab.edu

## Abstract

BK polyomavirus (BKPyV) is a major cause of kidney transplant failure, for which there are no antivirals. The current model is that BKPyV expresses TAg (large T antigen) early during infection, promoting cells to enter S phase where the viral DNA can access the host replication machinery. Here, we performed a single-cell analysis of viral TAg expression throughout the cell cycle to reveal that robust TAg expression required replication of the host DNA first. By using inhibitors that only affect host and not viral replication, we show that both TAg expression and viral production rely on an initial S phase. BKPyV is known to promote cellular re-replication, where the cell re-enters S phase from G2 phase (without passing through mitosis or G1 phase) to prolong S phase for viral replication. Thus, BKPyV infection results in cells with greater than 4N DNA content. We found that these subsequent rounds of replication of the host DNA relied on canonical host cell cycle machinery and regulators despite BKPyV infection. Together, these findings suggest a model for polyomavirus replication, where robust viral TAg expression depends on an initial host S phase and that BKPyV primarily replicates during host re-replication. Having a better understanding of the molecular events that are required for BKPyV production will help identify effective therapeutic targets against BKPyV.

## Author summary

BK polyomavirus (BKPyV) is the leading cause of graft loss in kidney transplant patients due to the lack of effective antivirals. Previous studies have implied that early expression of a viral protein, large T antigen (TAg), promotes S phase entry as an essential first step in viral replication. Using single-cell analysis of BKPyV infected kidney cells, we show that viral protein expression relied on an initial host S phase since robust expression of TAg was found only in cells already in S phase. Additionally, blocking the initial host S phase prevented viral protein expression and decreased viral production. Once viral proteins were highly expressed, however, viral production no longer depended on host

been deposited on Zenodo and can be access with the following link: https://doi.org/10.5281/zenodo. 13881804. Tools for analyzing the single-cell immunofluorescent data can be run using open-source ImageJ (NIH) and R packages and are provided at the following link https://github.com/SThompsonLab/MicroCyte.

**Funding:** This work was supported by the NIH (R01AI123162 and R21AI178734 to S.R.T) and the training program in CMDB T32 (GM008111 to J.M. N) led by Bradley Yoder at the University of Alabama at Birmingham. The funders had no role in study design, data collection and analysis, decision to publish, or preparation of the manuscript.

**Competing interests:** The authors have declared that no competing interests exist.

replication. Our findings support a new model where BKPyV initiates cell cycle entry independent of TAg since it was only expressed after an initial S phase. Under our new model, antiviral strategies could be developed to target BKPyV's dependence on the host cell cycle. Using these strategies will improve our understanding for how the virus replicates, enabling us to take a targeted approach for developing antivirals, and improve outcomes for kidney transplant recipients worldwide.

## Introduction

BK polyomavirus (BKPyV) is a small, double-stranded DNA virus that persistently infects the kidneys of approximately 180% of adults worldwide [1,2]. Although BKPyV reactivation is asymptomatic in healthy individuals, it causes graft loss in kidney transplant patients and is associated with increased risk of bladder cancer [3–5]. Despite the prevalence of BKPyV and the severity of complications, there are currently no specific treatments for BKPyV reactivation [6,7]. While BKPyV relies on the host replication machinery to replicate the viral DNA, it provides its own helicase, large tumor antigen (TAg)[1]. BKPyV replication using TAg and host replication machinery activates a cascade of kinases known as the DNA damage response (DDR), which prolongs S phase and is necessary for robust viral replication [8,9]. Following viral replication, BKPyV expresses the structural proteins VP1, VP2, and VP3, which encapsidate newly synthesized genomes to produce infectious virions [1,10]. While other DNA tumor viruses prevent host DNA replication through DDR activation, BKPyV uses DDR activation to induce and maintain host re-replication [8,11,12]. Importantly, the re-replication phenotype of the host DNA is also found in kidney biopsies from transplant patients with uncontrolled BKPyV reactivation, in which enormous, TAg+ nuclei are observed [13]. Given this requirement of host re-replication and the evidence of re-replication *in vivo*, a mechanistic understanding of how BKPyV induces re-replication is required.

Human DNA replication is temporally separated into two steps: origin licensing and origin firing [14]. During origin licensing in M and G1, the hexameric helicase components, MCM2-7, are recruited in a sequence-independent manner to origin recognition complexes (ORC) bound to the cellular DNA [14,15]. MCM licensing is tightly regulated by both transcription and protein degradation to prevent re-licensing of newly synthesized DNA in S phase; ensuring that DNA replication occurs only once per cell cycle. Tight regulation of re-licensing is essential as failure to prevent origin licensing in S phase results in irreparable DNA damage [16,17]. MCM activation, or firing, requires two kinases, Cdc7 and Cdk2, to recruit Cdc45 and the GINS complex and form the functional DNA helicase [18,19]. However, it has been shown that both Cdc7 and Cdk2 are inhibited by DDR activation [20,21]. Therefore, it is unknown how BKPyV induces re-replication without causing increased DNA damage [8], or how the virus blocks mitosis by activating the DDR without also inhibiting Cdc7 and Cdk2 [21]. To understand how BKPyV manages to replicate host DNA despite DDR activation, we used single-cell techniques to characterize the cell cycle during a BKPyV infection of human primary renal kidney epithelial (RPTE) cells, which are the site of natural infection in humans.

Our analysis of the cell cycle in BKPyV infected RPTE cells revealed that robust expression of TAg was primarily observed in the re-replicating population, which suggested that robust TAg expression depended on an initial host S phase. Indeed, inhibition of host origin licensing or firing prior to re-replication decreased viral titers and prevented TAg expression. Using dual-pulse labeling to track an initial S phase population throughout the cell cycle revealed a rapid accumulation of TAg as cells entered G2, which was followed by VP2/3 expression. Since

BKPyV required host replication, we investigated if BKPyV infection caused aberrant host licensing, but found that licensing pathways behaved canonically. Similarly, both host origin firing kinases Cdc7 and Cdk2 were required for efficient re-replication. Together, these data inform a revised BKPyV life cycle, in which BKPyV requires an initial host S phase for robust TAg expression and viral replication during host re-replication.

## Results

### Robust expression of TAg was observed only in the re-replicating population

DNA viruses often induce expression of host S phase proteins for viral replication [11]. Importantly, induction of S phase is typically coupled with cell cycle arrest to block concurrent host replication: decreasing competition with the host cell for replication resources. For example, adenoviruses and papillomaviruses arrest cells in G2 by blocking mitosis [22,23], while herpesviruses arrest cells at the G1/S boundary [24]. BKPyV infection is unusual in that the host DNA is re-replicated and maintained re-replication is required for viral replication [8]. To achieve induction into S phase and subsequent re-replication, the current model for BKPyV infection (Fig 1A) supposes early TAg expression first causes S phase entry for viral replication. Since viral replication activates the DNA damage response, mitosis is blocked and the cell undergoes re-replication. Importantly, loss of the re-replicating population drastically reduces viral titers even if the number of initial S phase cells is maintained [8]. To analyze the cell cycle profiles of mock and BKPyV infected RPTEs, we used a combination of 5-ethynyl-2'-deoxyuridine (EdU) pulse labeling and DNA content to generate single-cell cell cycle plots (Fig 1B). Cells were fixed and imaged at 24, 36, 48, and 72hpi, which revealed that there was no difference in the cell cycle profile between the mock and BKPyV-infected cells at 24hpi (Fig 1B and 1C). Later timepoints, however, showed an increase in S phase in the BKPyV-infected cells, which was largely attributed to an increase in re-replicating cells (Fig 1B and 1C) as demonstrated by the significant increase in the DNA content (mean DNA intensity) of the S phase cells (Fig 1D). Given the small size of the viral genome (5.2Kbp) relative to the human genome (>6Gbp) and the viral genome copy number per cell, viral replication could only account for <1% of the cellular DNA and, therefore, the observed increase in DNA content must be the result of host DNA re-replication [1,25]. Consistent with what others have observed, TAg expression was higher in the re-replicating cells (Fig 1E) [8,26,27]. Surprisingly, TAg expression was undetectable in the 2N population by fluorescent microscopy, which was unexpected as TAg expression is thought to be occur early to drive cells into S phase. The single-cell data, however, showed a 100-fold increase in TAg intensity as cells transitioned from the initial S phase to the 4N population (Fig 1F), while cells that were not yet in S phase had undetectable TAg levels (Fig 1E and 1F). Expression of the late genes revealed that VP2/3 expression followed the robust TAg expression as cells progressed to re-replication (Fig 1G), suggesting that infectious virions are formed in re-replicating cells. This agrees with the current model of the BKPyV life cycle, wherein robust expression of TAg precedes expression of the late proteins, but places viral replication after the cell arrived at 4N. Since even at early timepoints, robust TAg expression was consistently observed in ≥4N cells, these data suggest that BKPyV production is dependent on an initial S phase.

### Disrupting the initial S phase decreased BKPyV production

To determine if viral replication depended on an initial host S phase, we used chemical inhibitors to specifically prevent cellular DNA replication. Although cellular and viral DNA

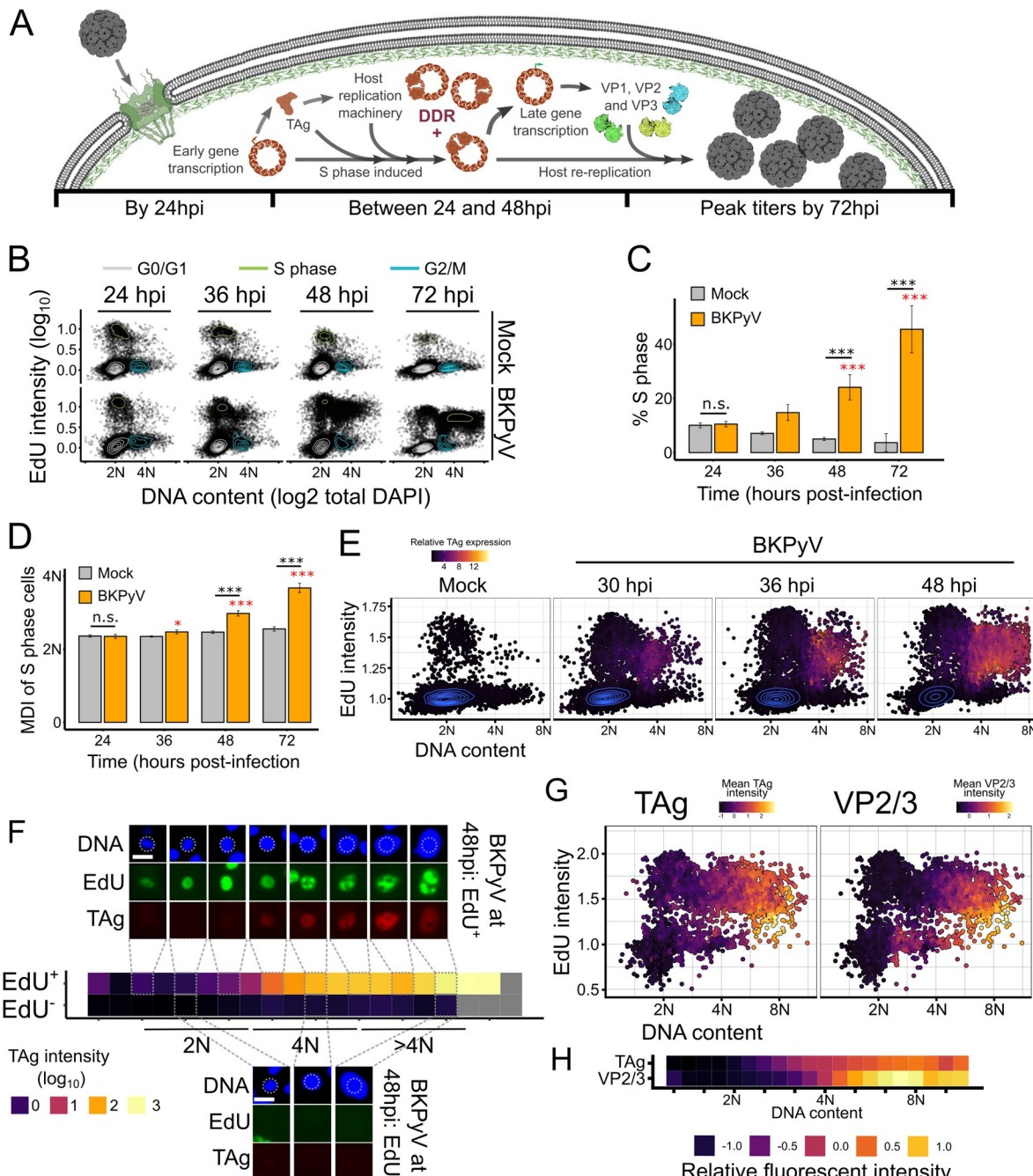

**Fig 1. Robust expression of TAg was observed only in the re-replicating population.** (A) Model of the current paradigm of a BKPyV infection. (B) Cell cycle diagrams generated from immunofluorescent assays (IFA) of mock or BKPyV-infected RPTE cells (MOI = 0.5) at 24, 36, 48, or 72hpi. Representative of n = 3 biological replicates is shown. (C) Quantification of cells in S phase from (A). Significance relative to mock or BKPyV at 24hpi are indicated by black or red asterisks, respectively. (D) Mean DNA intensity (MDI) of the S phase cells with significance as in (C). (E) Representative single-cell cell cycle diagrams (n = 3) generated from IFA images of mock (36hpi) or BKPyV-infected (MOI = 0.5; at 30, 36, or 48hpi) RPTE cells, pseudocolored by mean nuclear TAg intensity. (F) Representative binned analysis (n = 3) of TAg at 48hpi with randomly selected cells from each bin shown. For each replicate, 20 images were taken of each condition for an average of 18,000 cells/condition. Scale bar = 20 μm. (G) Representative cell cycle plots (n = 3) were colored based on TAg or VP2/3 intensity at 72hpi. (H) Representative binned analysis (n = 3) of TAg and VP2/3 expression at 72hpi. For each replicate, 20 images were taken of each condition for an average of 18,000 cells/condition. Statistical significance was determined by multi-factorial ANOVA and a Tukey post-hoc (*: p < 0.05, **: p < 0.01, ***: p < 0.001).

replication use similar replication proteins, BKPyV uses TAg as the viral helicase whereas cellular DNA uses the CMG (CD45, MCM2-7, GINS) helicase in a highly regulated, sequential manner [14,28]. Thus, interfering with host origin licensing should block host replication without directly affecting viral replication. Host origin licensing is carefully regulated by sequential activation of two E3 ubiquitin ligases, APC$^{Cdh1}$ and SCF$^{Skp2}$ [14]. APC$^{Cdh1}$ must be inhibited in late G1 to allow for origin licensing, and for the subsequent activation of SCF$^{Skp2}$ to degrade cell cycle inhibitors [29–31]. To determine the effect of disrupting host origin licensing on viral replication, APC, SCF, or both inhibitors were either added prior to S phase induction (early, 18hpi) or after the onset of host re-replication (late, 48hpi). Consistent with APC needing to be inactivated during host S phase and BKPyV requiring S phase proteins, viral titers were unchanged with either early or late inhibition of APC (Fig 2A). In contrast, early inhibition of SCF, but not late, reduced viral titers; consistent with the known role of SCF$^{Skp2}$ for S phase induction and that BKPyV therefore requires an initial S phase. Furthermore, addition of both inhibitors phenocopied SCF inhibition alone. Since neither SCF inhibition nor the dual inhibition decreased viral titers after the onset of host re-replication, this suggests that BKPyV required SCF$^{Skp2}$ activity early for viral production, but not after the onset of re-replication. Since SCF activity is required to degrade cell cycle inhibitors that may block viral replication, we also inhibited licensing by using siRNAs to knockdown MCM2-7 prior to a BKPyV infection. MCM knockdown decreased S phase induction following BKPyV infection and subsequent viral titers (S1 Fig). Since BKPyV is not thought to require MCMs as it encodes its own helicase [1], this supports the requirement of an initial S phase for a productive BKPyV infection as host S phase was decreased with MCM knockdown.

Following licensing, host DNA replication requires two kinases, Cdc7 and Cdk2, to fire the host origins [18,32–34]. Since TAg is not known to require either kinase to activate its helicase activity [35], inhibiting these kinases should prevent host, but not viral, replication. As with SCF inhibition, early inhibition of Cdc7, Cdk2, or both significantly decreased viral titers (Fig 2B). Furthermore, since Cdc7 activity is required prior to Cdk2 for origin firing, the dual inhibition phenocopied the Cdc7i as expected. Like what was seen with SCFi, late inhibition of either kinase did not decrease viral titers. Together, these data show that BKPyV requires an initial host S phase for viral production.

## Robust expression of TAg occurs following an initial S phase

Since robust levels of TAg were only observed in S phase cells that had ≥4N DNA content (Fig 1E and 1F) and suppression of host replication prior to re-replication decreased BKPyV production (Fig 2A and 2B), this suggested that BKPyV robustly expresses TAg following the initial S phase. To determine when TAg was expressed during the cell cycle, a pulse-chase time course was used to track an initial S phase population over time (Fig 2C). Mock or BKPyV-infected RPTEs were labeled with bromodeoxyuridine (BrdU) from 27-30hpi to identify cells in S phase during this time. The BrdU was washed out and different samples were labeled with EdU at various times (30-33hpi, 33-36hpi, 36-39hpi, 39-42hpi, 42-45hpi, and 45-48hpi). Thus, the BrdU+ population was tracked as it progressed through the cell cycle (Fig 2D). As expected, the mock BrdU+ population completed S phase, entered G2, and underwent mitosis (Fig 2D top, 30-39hpi). By 42hpi, some mock cells returned to S phase from the G1 population, although most remained in G1. BKPyV-infected cells followed similar kinetics with cells entering into a G2-like state around 33-36hpi. At 36hpi, however, the BrdU+ cells bifurcated as the TAg- cells returned to G1 while TAg+ cells remained at 4N (Fig 2D, bottom). Interestingly, while the G1 cells from both the mock and BKPyV-infected cells returned to S phase at a similar time (42hpi), the BKPyV-infected G1 cells returned to S phase at a higher frequency than

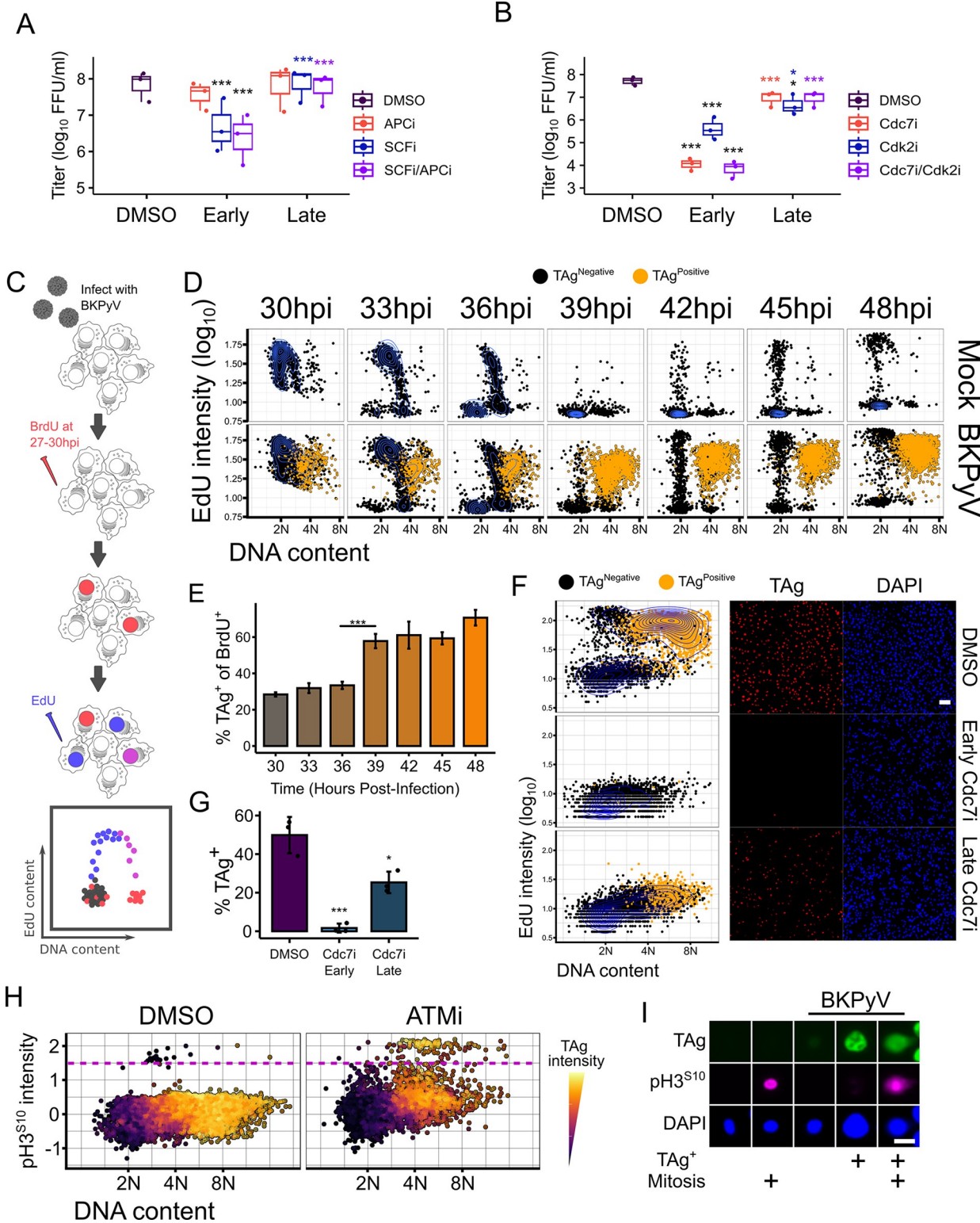

**Fig 2. BKPyV robustly expresses TAg in late S/G2 following the initial S phase.** (A-B) Viral titers (n = 3) from RPTEs treated with DMSO, (A) E3 ubiquitin ligases inhibitors (APCi: 10μM ProTAME and SCFi: 10μM MLN4924), or (B) origin firing inhibitors (Cdc7: 10μM PHA-767491, Cdk2: 3μM NU-6140). Inhibitors were added at 18hpi (Early) or at 48 hpi (Late) and virus was titered at 72hpi by focus forming assay. Black asterisks denote significance from the DMSO control while colored asterisks in the late groups denote significance from the early sample of the same inhibitor. (C) Experimental design for the BrdU/EdU pulse-chase labeling. Cells were labeled with BrdU from 27-30hpi, washed and grown

in label-free medium, then labeled with EdU for 3 hours prior to fixation at the indicated time points. (D) Cell cycle diagrams were generated for BrdU+ cells based on S phase (EdU) and DNA content (FxCycle Violet). Cells were gated based on nuclear TAg intensity and TAg+ cells are indicated in orange. Shown are representative cell cycle diagrams (n = 5) generated from IFA images of BrdU/EdU pulsed, mock or BKPyV-infected (MOI = 0.5) RPTEs. (E) Quantification of the percent of TAg+ cells in the BKPyV-infected, BrdU+ population from (D). (F) Representative cell cycle diagrams and IFA images (n = 3) of BKPyV-infected RPTE cells treated with either DMSO or the Cdc7 inhibitor at 18hpi (Early) or at 48hpi (Late). Cells were fixed at 72hpi and IFA was performed for cell cycle analysis and TAg intensity. Scale bar represents 100μm. (G) Quantification of percent TAg+ cells from Cdc7 inhibited cells from (F). (H) Representative plot (n = 3) of DNA content by mitosis (pH3S10) in BKPyV-infected RPTEs treated with the ATM inhibitor (10 μM, AZD0156) at 24hpi and fixed at 72hpi. Dots are colored to show TAg intensity and the dotted line indicates the cut-off used for mitotic cells. For each condition and replicate, at least 2,300 cells were imaged for analysis. (I) Representative images (n = 5) of mock or BKPyV-infected cells showing TAg and mitosis negative or positive cells. Scale bar represents 20μm. Statistical significance was determined using a one-way ANOVA and a Tukey post-hoc (*: $p < 0.05$, ***: $p < 0.001$). See also S1 and S2 Figs.

the uninfected cells (Fig 2D). Although BKPyV infected cells followed similar cell cycle kinetics as the mock infected cells, the TAg- BKPyV infected cells had a higher likelihood of re-entering S phase. Quantification of the TAg+ cells within the BrdU+ population at each timepoint revealed that, while the percent of TAg+ cells remained unchanged between 30 and 36hpi, there was a rapid increase in the number of TAg+ cells at 39hpi corresponding with an exit from S phase and entering G2 (Fig 2D and 2E). After the rapid increase in TAg+ cells, the percent of TAg+ cells within the BrdU+ cells plateaued as cells that underwent mitosis did not go on to express TAg. These data show that BKPyV-infected cells robustly express TAg after the initial host S phase and near the onset of re-replication, while cells that did not express TAg underwent mitosis.

Since robust TAg was only expressed after S phase entry, this suggested that the decrease in titers with early S phase inhibition (Figs 2A and 2B and S1) was due to impaired TAg expression. To test this, the origin firing kinase Cdc7 was inhibited in BKPyV-infected RPTEs either prior to the initial S phase (early, 18hpi) or after the initial S phase (late, 48hpi) and the cell cycle state and TAg expression was determined by IFA at 72hpi. As expected, Cdc7i at either timepoint dramatically decreased the amount of S phase (Fig 2F). Furthermore, cells inhibited early displayed very few TAg+ cells (Fig 2F and 2G), while late Cdc7i resulted in robust TAg expression limited to the $\geq$4N population. While there was a reduction in TAg+ cells with the late treatment compared to the DMSO control (Fig 2G), the slight reduction was likely due to the loss of cells that would have begun S phase between 48 and 72hpi and, therefore, failed to reach late S/G2 with late Cdc7i (Fig 1B and 1C). Since the Cdc7 inhibitor may have off-target effect such as inhibiting transcriptional Cdks, the host MCM helicase was inhibited with heliquinomycin using the same early/late treatment (S2 Fig). This resulted in a similar loss of TAg expression with early, but not late, MCMi despite being structurally and mechanistically different from the Cdc7 inhibitor. Since the BrdU/EdU pulse data indicated that TAg was robustly expressed after the initial S phase and the inhibition of an initial S phase prevented robust TAg expression, these findings support a model in which robust TAg expression is dependent upon host S phase.

## Robust TAg expression occurs in G2 prior to re-replication

While robust TAg expression required an initial S phase, we could not determine if the increase in TAg occurred before or after the onset of re-replication. Previously, we reported that ATM inhibition during a BKPyV-infection caused cells to complete S phase and enter mitosis rather than re-replicating [8]. Therefore, the TAg state of the mitotic cells during ATMi should reveal whether TAg is expressed robustly before re-replication or after. In agreement with our previous findings, ATMi increased the number of mitotic cells at 4N (Fig 2H). Importantly, the overwhelming majority of these mitotic cells displayed robust TAg intensity (Fig 2H and 2I), suggesting that TAg expression preceded re-replication. These data show that

the robust expression of TAg during a BKPyV infection occurs after the initial S phase induction, but prior to the onset of re-replication. Thus, robust TAg is expressed in late S or G2.

## TAg expression in G2 is not dependent on genome copy number or protein degradation

One reason that an initial S phase may be essential is that a threshold of viral genomes is required for robust TAg expression. Under this model, the threshold must be reached by viral replication during the initial S phase or robust TAg expression is not achieved. A requirement for viral genome amplification in order for robust TAg expression would explain our pulse-chase analysis (Fig 2D) where the subset of cells that failed to become TAg+ underwent mitosis. Thus, we infected RPTE cells with increasing amounts of virus (MOI = 0.01 to 40), and quantified TAg expression and cell cycle states at 48hpi to determine if high genome numbers decoupled TAg expression from re-replication. While the percent of S phase cells did generally correlate with increasing MOIs, it did not increase linearly past an MOI of 0.5 and, even at an MOI of 40, there was never more than ~75% S phase (Fig 3A). To see if these super-infected, non-S phase cells were expressing TAg, we quantified the percent of TAg+ cells (Fig 3B). Similar to the levels of S phase, there was a positive, nonlinear correlation between the MOI and percent TAg+, and we were unable to achieve more than ~75% of TAg+ cells (Fig 3B). The observation that ~25% of RPTEs did not express robust TAg even at a high MOI suggested these cells may be resistant to infection (Fig 3B and 3C). Since TAg expression required an initial S phase, we hypothesized that the cell cycle of TAg- cells would have very little S phase. Unexpectedly, the TAg- cells displayed higher rates of S phase than mock cells (Fig 3D), but did not undergo re-replication (Fig 3E). One possibility is that these cells are infected and BKPyV is driving S phase induction, but TAg fails to be robustly expressed for unknown reasons. Alternatively, these cells may represent a distinct sub-population that is not permissive to TAg expression, but naturally replicate at a higher rate than the other RPTEs. Regardless, analysis of the TAg+ cells showed that the cell cycle distribution of TAg+ cells was largely insensitive to MOI (Fig 3F and 3G), as TAg+ cells were primarily found only in late S/G2 cells even at high MOIs.

Another possibility is that TAg expression is regulated by degradation, as has been observed in Merkel cell polyomavirus [36]. Under this model, BKPyV TAg would be constitutively expressed, but degraded outside of G2 and preventing sufficient accumulation for detection by IFA. Indeed, the activities of multiple E3 ubiquitin ligases are known to shift throughout G2 and mitosis [14]. To test if TAg protein levels were subject to degradation, we transiently inhibited the proteasome in mock and BKPyV infected RPTEs from 48 to 54hpi and quantified TAg levels by western analysis (Fig 3H and 3I). Inhibition of the proteasome did not increase TAg levels, suggesting BKPyV TAg was not being actively degraded. As a positive control, proteasome inhibition increased the abundance of Cdk2 (Fig 3H), which is upregulated during a BKPyV infection and degraded throughout the cell cycle [37,38]. In fact, we observed a mild, but significant, decrease in TAg levels (Fig 3I), which may be due to the requirement of protein degradation for S phase induction.

## MCMs are relicensed for re-replication during BKPyV infection

Given BKPyV's requirement for an initial S phase and induction of re-replication, we next looked at host replication pathways. Licensing the origins of replication throughout G1 is dependent on both transcription and the activity of two E3 ubiquitin ligases APC$^{Cdh1}$ and SCF$^{Skp2}$, which control the levels of licensing factors Cdc6 and Cdt1, respectively, for MCM2-7 recruitment onto the DNA (Fig 4A) [14]. For cells to replicate past 4N during BKPyV

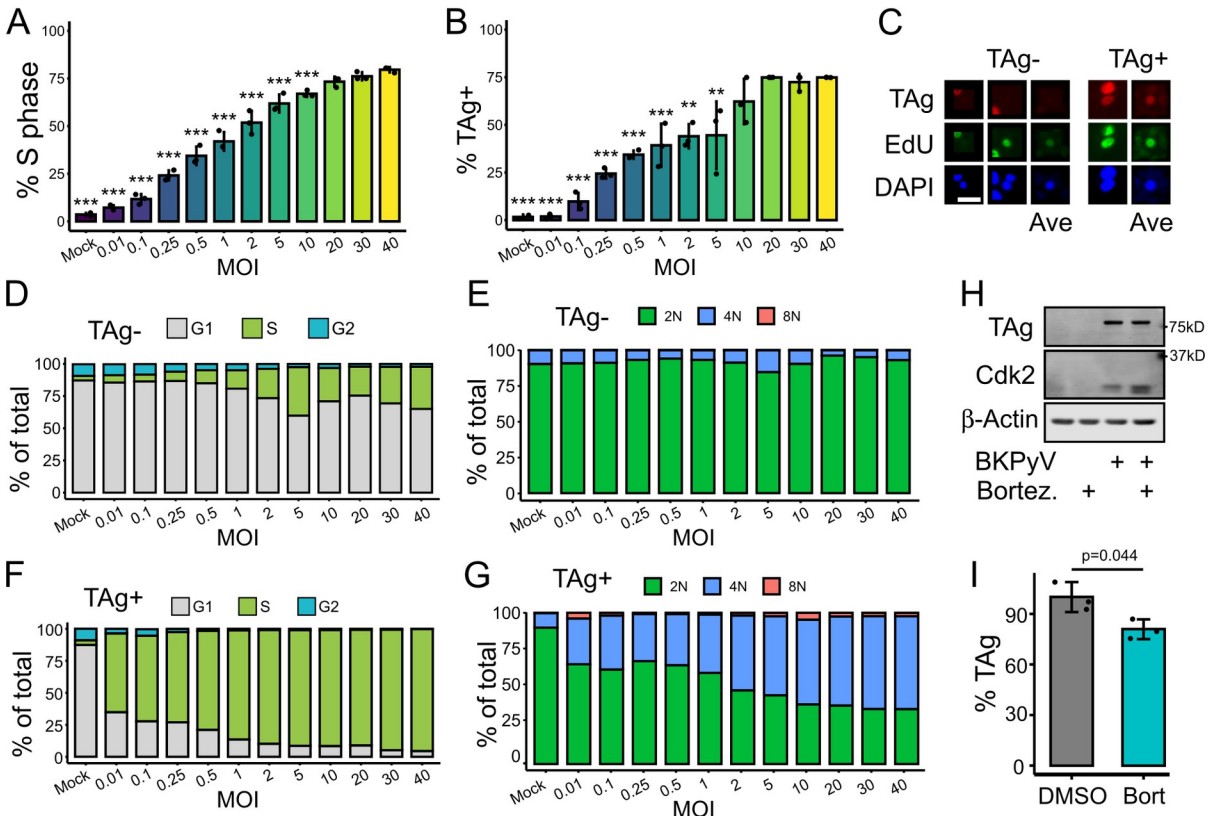

**Fig 3. Robust TAg expression is not regulated by genome copy number nor degradation.** (A-G) RPTEs were infected with increasing amounts of BKPyV (MOI = 0.01 to 40) and fixed at 48hpi, imaged, and used to quantify percent S phase (A-C) and percent TAg+ (B) with representative images of cells from MOI = 40 shown in (C). To generate an averaged image, pixel intensities of 20 cells randomly selected from the TAg- and TAg+ populations were averaged to create a new image. Statistical significance was determined by one-way ANOVA using a Tukey post-hoc with asterisks denoting significance from the 40 MOI sample (n = 3; **: $p < 0.01$, ***: $p < 0.001$). Scale bar represents 20μm. (D, E) The TAg- populations were gated into G1, S, and G2 populations (D) to visualize cell cycle distribution, or into 2N, 4N, and 8N populations to visualize re-replication (E). (F, G) The same gating analysis as shown in (D,E) was used for the TAg+ population. (H, I) Representative western analysis (H) and TAg quantification (I) (n = 3) of mock or BKPyV-infected RPTE cell lysates treated with either DMSO or the proteasome inhibitor bortezomib (0.5μM) and collected at 48hpi. Statistical significance was determined using a Student's t-test (H).

infection, either new origins of replication must be licensed on the DNA following the initial S phase or BKPyV induces re-replication using a noncanonical helicase. To determine if origins were being relicensed with MCMs, BKPyV-infected RPTEs at 48hpi were stained for total and chromatin-associated MCM4. Since MCMs are expressed at a much higher abundance than is used for origin licensing, permeabilization of cells prior to fixation was used to wash out (W/O) unbound MCM4 and distinguish between total MCM4 and chromatin-bound MCM4 as part of the origin complex [39]. As expected, MCM4 intensity was high in G1 and early S populations for both the total and chromatin-associated (W/O) mock and BKPyV infected RPTEs (Fig 4B and 4C). In both mock and BKPyV-infected cells, the chromatin-associated MCM4 intensity rapidly decreased during S phase as unfired origins became disassembled, and was fainter in G2 cells (Fig 4B and 4C). In the >4N population of BKPyV-infected cells, there was a prominent chromatin-associated MCM4+ population concurrent with re-replication (See 4N EdU+, Fig 4B–4D). The increase in chromatin-associated MCM4 following the initial S phase suggests that MCM re-licensing occurs during BKPyV infection and that the canonical CMG helicase is used for BKPyV-induced re-replication.

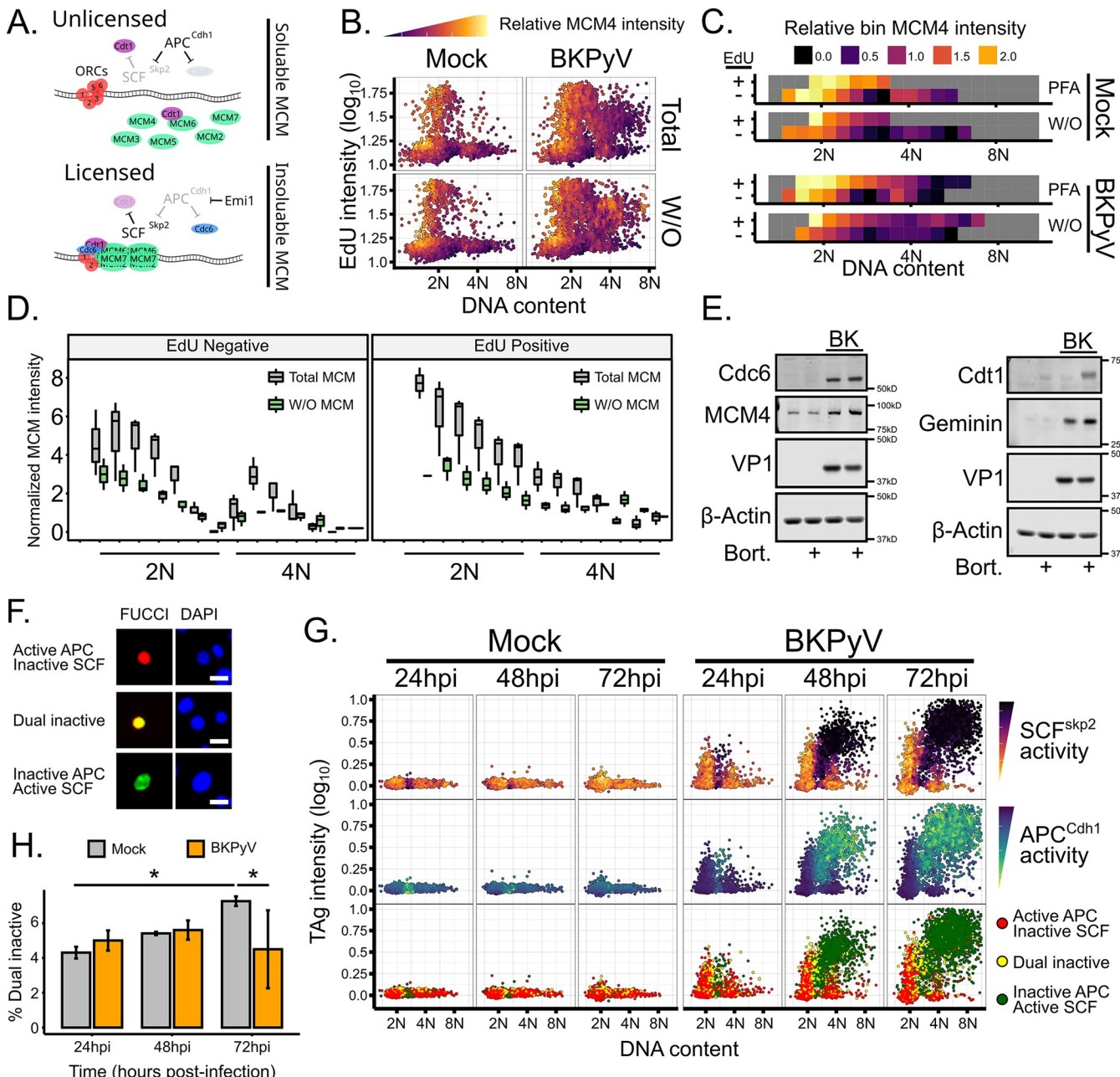

**Fig 4. Origin licensing uses canonical mechanisms during BKPyV-induced re-replication.** (A) Diagram of host origin licensing and MCM solubility. (B-D) Representative cell cycle plots from mock and BKPyV-infected RPTEs at 48hpi and colored by relative MCM4 intensity. Total MCM4 was visualized by fixing cells in PFA, while chromatin-associated MCM4 (W/O) was visualized by permeabilizing cells prior to fixation. (C) Representative binning analysis of PFA-fixed (Total) or washout (W/O) data shown in (B) was split by EdU status and binned by DNA content. (D) Quantification of the binned data represented in (C). (E) Representative western analyses of licensing factors after proteasome inhibition (Bort.) at 48hpi. (F-H) Representative images of FastFUCCI transduced RPTEs (F). Scale bar represents 20μm. (G) Representative plot of FastFUCCI-transduced mock or BKPyV-infected (MOI = 0.5) cells fixed at 24, 48, and 72hpi. Only cells positive for a FastFUCCI fluorophore are represented and colored based on fluorophore intensities. (H) Quantification of dual positive cells shown in panel G. Statistical significance was determined by two-way ANOVA using infection state and time and a Tukey post-hoc (n = 3; *: p < 0.05). See also S3 Fig.

## APC$^{Cdh1}$ and SCF$^{Skp2}$ follow canonical activation patterns during BKPyV infection

Since host relicensing outside of G1 would require Cdt1, SCF$^{Skp2}$ would need to be inhibited for Cdt1 levels to increase and allow host origin re-licensing to occur (Fig 4A). Therefore, BKPyV infection may specifically block SCF$^{Skp2}$ activity to promote host re-licensing through Cdt1 upregulation [40]. Indeed, the manipulation of E3 ubiquitin ligases has been well-established for other DNA viruses [41,42] and would allow BKPyV to efficiently induce re-replication. To determine if the licensing factors were being actively degraded, the proteasome was inhibited during re-replication for 6 hours in mock and BKPyV infected RPTEs (Fig 4E). Consistent with our earlier findings that suggested APC was not active in BKPyV infected cells (Fig 2A), none of the APC targets (Cdc6, geminin, and MCM4) increased when the proteasome was inhibited, suggesting that APC$^{Cdh1}$ activity is not increased with BKPyV infection. In contrast, Cdt1 increased with proteasome inhibition in both mock and BKPyV-infected cells, suggesting that SCF$^{Skp2}$ was active and Cdt1 was degraded during a BKPyV infection. It is possible, however, that aberrant SCF$^{Skp2}$ inhibition occurred only in a subset of cells and, thus, transient Cdt1 stability was undetectable by western analysis. To directly measure APC$^{Cdh1}$ and SCF$^{Skp2}$ activity, we performed a single-cell analysis using the FastFUCCI reporter to assay the activity of the APC$^{Cdh1}$ and SCF$^{Skp2}$ across DNA content in mock and BKPyV-infected cells [43]. The FastFUCCI reporter expresses two fluorescent proteins fused to the degrons of either Geminin or Cdt1, which are targeted by APC$^{Cdh1}$ and SCF$^{Skp2}$, respectively. Since both these fluorophores are expressed from the same mRNA, the fluorescent intensity is a direct readout of APC and SCF activity (Fig 4F). Since we were only able to achieve ~20% transduction of the FastFUCCI lentivirus in primary RPTEs (S3A Fig), analysis was restricted to cells with detectable FastFUCCI fluorescence (Fig 4G and 4H). Importantly, there was no difference in transduction between mock and BKPyV-infected cells (S3A Fig), nor was there a difference in viral production with FastFUCCI transduction (S3B Fig). As expected of uninfected cells, APC$^{Cdh1}$ activity was observed in the 2N and 4N population while SCF$^{Skp2}$ activity was restricted to the S phase zone between 2N and 4N (Fig 4G, left). While a few cells displayed dual-inactivation, these made up 4–6% of the population and were found in the 2N population where dual inactivation in G1 is required to complete origin licensing [14] (Fig 4H). BKPyV-infected cells displayed a similar distribution of APC and SCF activity as uninfected cells, with APC activity being restricted to the 2N and 4N populations and SCF activity in the S phase/re-replicating populations (Fig 4G). There was not an increase in the percent of dual inactivated cells upon BKPyV infection (Fig 4H), thus BKPyV infection did not appear to alter the activity distribution of the E3 ubiquitin ligases regulating licensing.

## BKPyV-induced re-replication increased canonical phosphorylation of the host helicase

Since DDR activation prevents the replication of damaged DNA by inhibiting origin firing and BKPyV robustly activates the DDR (Fig 5B), it is unclear how new origins fire in the context of DDR activation [20,21,44]. Origin firing requires the sequential phosphorylation of licensed MCM complexes by Cdc7 and Cdk2, which recruits the other components of the functional helicase, Cdc45 and the GINS complex, respectively (Fig 5A) [18,19]. To determine if canonical origin firing occurs during BKPyV-induced re-replication, phosphorylation of the Cdc7 target MCM2 at serine 53 was determined with and without a Cdc7 inhibitor (Fig 5C) [45]. Western analysis showed that BKPyV infection significantly increased the amount of pMCM2$^{S53}$ and Cdc7i reduced pMCM2$^{S53}$, indicating that phosphorylation of MCM2 was dependent on Cdc7 (Fig 5C). To test whether MCM2 phosphorylation occurred during re-

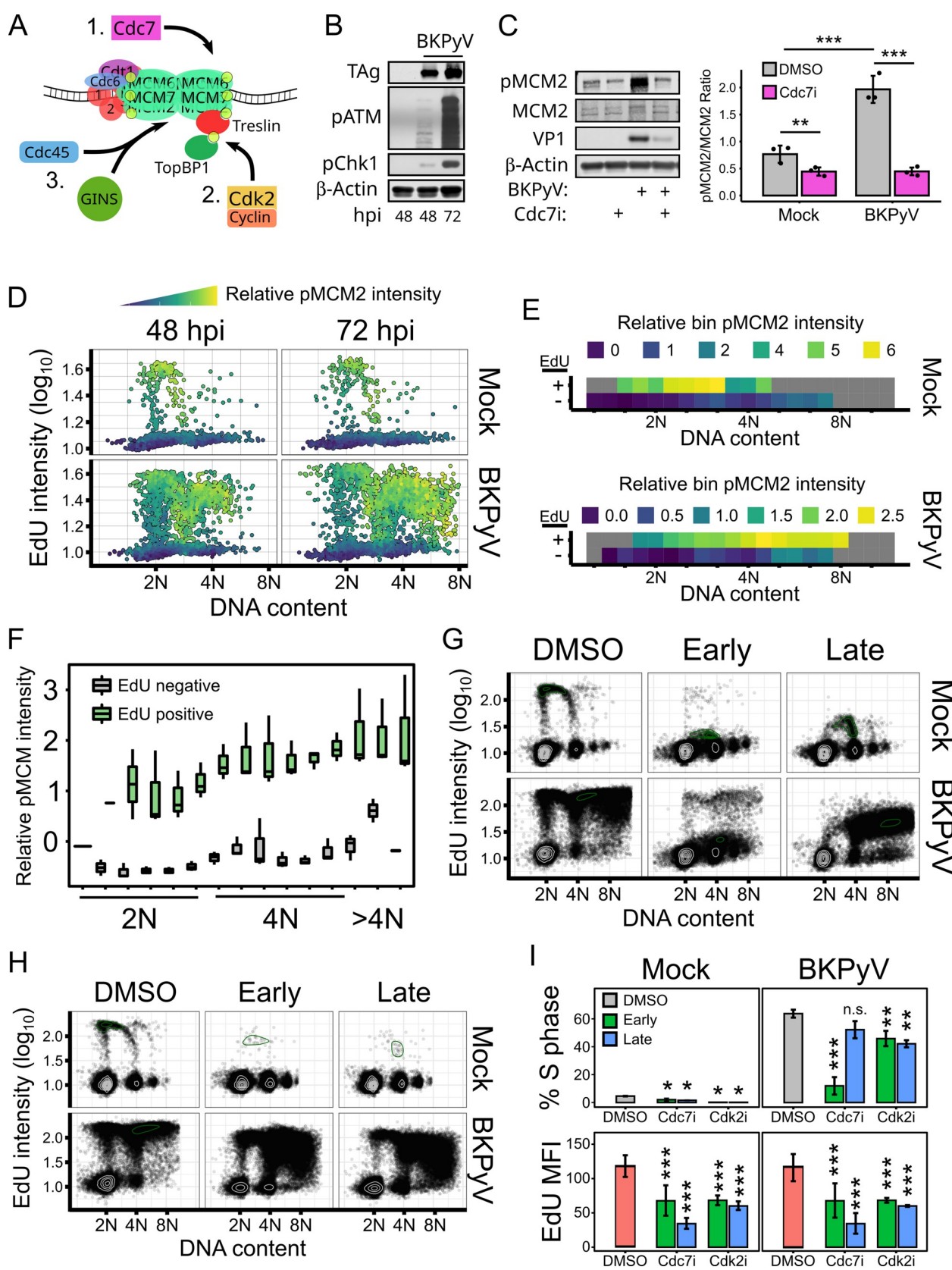

**Fig 5. Firing of new origins of replication during BKPyV-induced re-replication required canonical kinases Cdc7 and Cdk2.** (A) Diagram of canonical host origin firing. (B) Representative western analysis of viral proteins and markers of DDR activation from mock and BKPyV-infected RPTEs at various timepoints post-infection (n = 3). (C) Representative western analysis of mock and BKPyV-infected RPTEs treated with DMSO or Cdc7 inhibitor PHA-767491 (15μM) at 18hpi and collected at 48hpi with quantification (n = 3). (D-F) Representative cell cycle analysis of mock or BKPyV-infected RPTE cells fixed at 48 and 72hpi colored by relative pMCM2$^{S53}$ intensity (n = 3). Relative pMCM2$^{S53}$ intensity was calculated by dividing the pMCM2$^{S53}$ intensity of each cell by the sample average of pMCM2$^{S53}$. (E) Representative binning analysis of 72hpi cells shown in panel. Boxes are colored by relative pMCM2$^{S53}$ intensity, except for boxes with fewer than 10 cells (gray). (F) Quantification of the pMCM2$^{S53}$ intensity from the binning analysis in panel (E). (G-I) Flow cytometry-based cell cycle analyses at 72 hpi of mock and BKPyV-infected RPTEs treated at 18hpi (Early) or 48hpi (Late) using 15μM Cdc7 inhibitor PHA-767491 (G) or 10μM Cdk2 inhibitor NU-6140 (H). (I) Quantification of S phase and EdU mean fluorescent intensity (MFI) of the cells from data represented in (F) and (G). Statistical significance was determined by (C) two-way ANOVA using infection and Cdc7i or (I) one-way ANOVA, followed by a Tukey post-hoc with asterisks denoting significant differences from the DMSO sample (n = 3; *: p < 0.05, **: p < 0.01, ***: p < 0.001). See also S4 Fig.

replication, a single-cell cell cycle analysis was performed to determine the distribution of pMCM2$^{S53}$ in mock or BKPyV-infected cells at 48 and 72hpi (Fig 5D and 5E). As expected, the initial S phase in both mock and BKPyV-infected cells had high pMCM2$^{S53}$ levels (Fig 5D and 5E). In binned populations, MCM2 phosphorylation increased throughout the initial S phase, representing continual origin firing as S phase progressed. In BKPyV-infected cells, there was a further increase in pMCM2 in the 4N cells in S phase (EdU+) as they began re-replication (Fig 5D–5F), indicating that new origins were being fired.

While there was an increase in Cdc7-dependent MCM2 phosphorylation, this may not be necessary for host re-replication. Therefore, we used flow cytometry to analyze the requirement of Cdc7 (Fig 5G and 5I) and Cdk2 (Fig 5H and 5I) for BKPyV-induced re-replication by treating cells with either a Cdc7i or a Cdk2i at 18hpi (Early) or 48hpi (Late). Due to the greater range of flow cytometry compared to IFA, we were able to identify S phase populations with even low levels of EdU incorporation. As expected, both Cdc7i and Cdk2i in mock infected cells significantly decreased S phase at both early and late treatment times (Fig 5G–5I). Additionally, uninfected cells that were able to enter S phase despite Cdc7i or Cdk2i showed significantly reduced EdU incorporation (Fig 5I, bottom), suggesting a reduction of active replication complexes. While early Cdc7i in BKPyV-infected cells significantly reduced S phase and re-replication, late inhibition only mildly decreased S phase (Fig 5G–5I). However, the reduced EdU intensity with late Cdc7i in both the mock and BKPyV-infected cells indicate that S phase progression was impeded. Thus, Cdc7 was required for both the initial S phase and re-replication. Cdk2i also decreased S phase and EdU incorporation in BKPyV-infected cells, albeit to a lesser extent than early Cdc7i (Fig 5H and 5I). Given the recent findings that Cdk2 inhibition is rapidly supplemented by other cellular Cdks, it is likely that the lesser effect of Cdk2i is due to the functional redundancy of other Cdks [38]. These results indicated that both Cdc7 and Cdk2 were required for efficient S phase, which conflicts with the role of the DDR inhibiting these kinases. One possibility for how origin firing and DDR activation both occur during re-replication is that DDR activation and origin firing were restricted to different populations. Under this model, DDR activation during re-replication would not impede origin firing occurring in the preceding G2 population. Indeed, single-cell analysis of DDR activation (S4 Fig) showed that both ATR and ATM were primarily activated in the re-replicating cells, while cells in G2 (4N, EdU-) did not have the DDR activated above mock cells. Taken together, these results suggest that origin firing during a BKPyV infection uses the canonical kinases Cdc7 and Cdk2, and DDR activation does not prevent these kinases from firing origins as the newly licensed origins are fired prior to DDR activation.

## Discussion

Using single-cell techniques, our studies revealed that BK polyomavirus depends on the host cell cycle for TAg expression and viral production. Our data contrasts with the model where

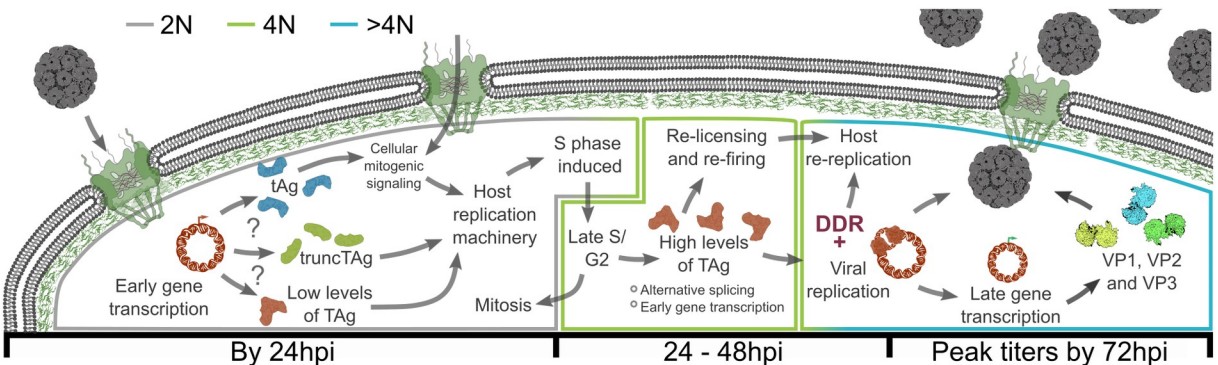

**Fig 6. Model for the dependence of BKPyV on host replication.** Robust S phase may be induced by 1) tAg enhanced mitogenic signaling through disruption of the protein phosphatases, 2) expression of the truncated TAg that contains a pRb binding domain, or 3) low levels of TAg that are primarily directed towards pRb inhibition. As the cell enters late S or G2, TAg rapidly accumulates and inhibits pRb to stimulate host re-replication through origin re-licensing and re-firing using canonical mechanisms. Since the rapid TAg accumulation in the late S or G2 populations cannot be explained by an increase in genomes, it may be due to a phase-specific shift in early gene splicing and/or transcription. In the re-replicating population, BKPyV uses host replication machinery and TAg to replicate viral genomes. High levels of viral replication here activate the DNA damage response (DDR), which maintains a re-replicating population. After TAg expression and viral replication, the structural proteins are expressed, and viral progeny are detected concurrent with host re-replication.

TAg is expressed early and pushes cells into the initial S phase for viral replication. Instead, our data supports a new model whereby BKPyV requires an initial host S phase to replicate, which aligns better with the other small DNA tumor viruses such as papilloma- and adenoviruses [23,46]. While our data agree that TAg expression is observed before the viral capsid proteins, the cells robustly expressing TAg were in late S/G2 rather than G1 even at early times post-infection. Additionally, inhibiting host replication did not result in the accumulation of TAg expression in the G1/early S phase population, but prevented robust TAg expression all together and reduced viral titers; demonstrating that TAg expression depends on an initial host S phase. If host replication was inhibited after infected cells had already reached the 4N DNA content and TAg was expressed, host replication was no longer required for maintaining TAg expression or viral production. Proteasome inhibition failed to increase TAg expression suggesting that the increase of BKPyV TAg in G2/M is not due to a shift in protein stability, as has been reported for the related MCPyV [36]. Furthermore, TAg expression was not the result of genome amplification during an initial S phase, since infecting with up to 80-fold more virions per cell still did not result in TAg+ cells in G1/early S phase. Since re-replication is rare in uninfected RPTEs and activation of the DDR is known to prevent DNA replication, we expected to find evidence of aberrant host origin licensing and firing. Instead, canonical origin licensing and firing mechanisms were required for re-replication and displayed an expected distribution of activity across the cell cycle. Together, these findings change our understanding of the BKPyV life cycle to include the requirement of an initial S phase for robust TAg expression and viral production (Fig 6).

Our new model shares similarities with other small DNA tumor viruses while accounting for a major difference in BKPyV, which expresses the viral helicase and cell cycle regulatory domains in a single protein. Many DNA tumor viruses encode proteins that promote S phase by inhibiting pRB (the S-phase E2F transcriptional repressor), unwind viral DNA for replication, and inhibit p53 (a DDR-responsive protein that can block DNA replication) [47]. While the polyomavirus TAg performs all three functions (pRb inhibition, DNA unwinding, and p53 inhibition), other DNA tumor viruses use different proteins to perform these functions. By using multiple proteins, this allows for differential expression at specific stages of the viral life cycle or host cell cycle [48,49]. While it was surprising that robust TAg expression depended

on a G2 environment, it is similar to papillomaviruses where the E1 helicase is robustly expressed in G2 [46]. Although, we cannot rule out the possibility that low levels of TAg are sufficient to promote S phase by inhibition of pRb, we posit that low TAg levels are unlikely to promote S phase given that the human genome contains 20,000 to 30,000 E2F1 binding sites [50]. Thus, low levels of TAg expression would be insufficient to bind up all the pRB for the initial S phase and replicate the viral genome. Another possibility is that an alternative splice variant drives an initial S phase and TAg is preferentially expressed only after reaching G2. Indeed, mRNA splicing regulation is known to vary across the cell cycle [51] and the TAg variant, truncated large T antigen (truncTAg), contains the pRb binding domain that could drive an initial S phase induction [52]. Since truncTAg does not possess a helicase domain, which is required for DDR activation [44], this may explain why we did not detect DDR activation until robust expression of TAg. Alternatively, the small t-antigen (tAg) splice variant binds to the phosphatase PP2a, which regulates mitogenic signaling through MAPK and may trigger cell cycle entry [53,54]. Indeed, tAg expression alone is sufficient to immortalize cells [55]. Finally, it may be that virus entry is sufficient to promote a burst of S phase induction. Indeed, MAPK inhibition during infection has been shown to decrease polyomavirus infection [56]. Furthermore, SV40 and JCPyV attachment alone is sufficient to promote MAPK activation [57]. While we did not find that BKPyV infection increased MAPK activation following attachment (S5 Fig) nor did we observe any difference in S phase induction until >24 hours post-infection (Fig 1), it is possible that a later step during infection, such as escape from the endoplasmic reticulum or nuclear entry, activates MAPK to drive the initial S phase prior to BKPyV transcription. Further studies will be required to determine whether the initial S phase is induced by expression of TAg splice variants.

While we have shown here that the cell cycle-specific accumulation of TAg is independent of proteasomal degradation and is insensitive to viral genome numbers, the mechanism underlying the rapid increase in TAg remains unclear. If TAg splicing is cell cycle dependent, it may be that the other splice variants, truncTAg and tAg, predominate until G2/M when TAg is preferentially spliced. Alternatively, BKPyV early gene transcription itself may depend on host transcription factors that are preferentially activated during late S phase, such as the MuvB/B-Myb (MMB) complex [58]. Interestingly, MuvB is also a component of the DREAM complex, which complexes with the pRb paralogs p107/p130 to repress cell cycle transcription [59]. TAg binding to the pRb family members may then serve multiple functions through increasing cell cycle entry by dismantling the DREAM complex, promoting S phase entry by binding pRb, and increasing G2/M transcription by promoting MMB formation. If the BKPyV early promoter recruited MMB complexes, this could result in a feedback loop to rapidly increase transcription of the TAg mRNA in late S/G2. Finally, there may exist cell cycle-specific changes in TAg translation that decouple TAg mRNA abundance from protein abundance. For example, noncanonical translation initiation has been reported to be altered across the cell cycle and facilitates mitotic progression [60]. Alternatively, TAg mRNA stability may be cell cycle dependent through an unknown mechanism.

BKPyV infection of primary cells induces cells to endoreduplicate, whereby the host genome is re-replicated without cells progressing through the cell cycle phases M and G1. Given that licensing of origins should not occur in S phase and origin firing should be inhibited by the DDR, we were similarly surprised to find that host origin licensing and firing appeared to behave canonically during BKPyV infection. However, our single-cell techniques resolved these discrepancies by identifying a G2-like state wherein licensing may occur outside of S phase and suggested temporal separation of origin firing with DDR activation. Since robust DDR activation occurred during re-replication, firing kinases Cdc7 and Cdk2 would not be inhibited for the induction of re-replication. While the timing of origin firing preceding

DDR activation explains how new origins of replication are licensed and fired during a BKPyV infection, it raises the question as to how an initial mitosis is blocked if DDR is not activated until after re-replication has begun. One possibility is that BKPyV activates the DDR kinase ATM to block mitosis, while the other kinase, ATR, is not activated until re-replication. This is supported by our previous findings that these kinases serve different roles during BKPyV infection with ATM promoting S phase entry and ATR preventing premature mitosis [8]. Indeed, papillomavirus E1 helicase is expressed in G2-arrested cells and can activate ATM without viral replication [61]. Alternatively, the initial mitotic arrest may be caused by another TAg splice variant independent of DDR activation. For example, although papillomavirus E1 can activate ATM, G2 arrest is caused by E4, which sequesters the mitosis promoting factor in the cytoplasm [22,62,63]. Another possibility is that the initial mitotic arrest is a cellular response to acute stress known as endocycling during which mitotic arrest and a burst of S phase transcription induces programmed re-replication [64]. Indeed, acute stress triggers murine kidney epithelial cells to naturally re-replicate independent of viral infection [65]. If TAg expression was responsive to an endocycling signal, BKPyV infection may skew RPTE replication towards endocycling rather than directly blocking mitosis. Alternatively, the sudden expression of TAg in G2 may mimic an endocycling signal in RPTEs by providing the rapid inhibition of pRb and the subsequent expression of new S phase proteins such that the inhibition of mitosis is caused by the onset of re-replication.

Taken together, our findings support a new model by which early S phase is induced during viral infection by either low levels of TAg expression or the expression of other TAg splice variants (Fig 6). Once the cells are in late S/G2 phase, TAg is robustly expressed to amplify the viral genome via its helicase domain. Since TAg also contains both a pRb binding domain and a p53 binding domain, robust TAg expression in the late S/G2 cells induces a secondary wave of S phase transcription and prevents p53-mediated cell cycle arrest. While further research is needed to understand the cell cycle dependent regulation of TAg, we posit that the rapid accumulation of TAg triggers a response similar to canonical endocycling. While RPTE cells can naturally endocycle in response to stress [65], the mechanisms underlying this stress response are unknown. Similarly, the mechanisms regulating BKPyV reactivation are unknown, as is much of the viral lifecycle *in vivo*. In other organisms, endocycling requires both S phase transcription in G2 and the suppression of mitosis, which is achieved by various mechanisms [64]. While TAg expression would promote S phase transcription, it is less clear how mitotic suppression is occurring as robust DDR activation was observed after the onset of re-replication. Indeed, it may be possible that BKPyV-induced re-replication represents an endogenous cellular response to infection that BKPyV has evolved to depend upon. This dependence of BKPyV on the host cell re-replication program may also explain the cellular tropism of BKPyV. While infection of renal endothelial cells by BKPyV activates an interferon response, infection of the RPTE cells does not and, interestingly, interferon signaling in response to RNA viruses is impaired during G2 [66,67]. Therefore, the propensity of RPTEs to endocycle may help explain why BKPyV naturally persists in these cells. Together, our study highlights the importance of both single-cell analyses on viral infection and dependence of BKPyV replication on the host cell cycle.

## Materials and methods

### Cell culture, viral infections, and chemical inhibition

Primary renal proximal epithelial (RPTE, Lonza #CC-2553) cells were grown in REGM (Lonza #CC-3190) at 37˚C in 5% $CO_2$ for 4 passages before being used in experiments. Primary RPTE were plated and grown to 80% confluency before infections. Viral infections were performed

by rocking for 1 hour at 4°C using 25% of the growth media inoculated with the viral stock (MOI = 0.5). Then, the infection media was removed, fresh REGM was added, and the cells were incubated at 37°C in 5% $CO_2$. Viral titers were determined by focus forming assay (FFA) in RPTE cells and quantified from 10 randomly selected fields of view (FOV). Viral stocks of BKPyV[Dunlop] were prepared by transfecting BKPyV genomes into HEK293TT cells using Lipofectamine 2000 (ThermoFisher 11668019), and grown in DMEM with 10% FBS and 2.5mM L-glutamine. After two weeks, the transfected RPTE cells were scraped into the supernatant, collected, and titered by FFA. Small molecule inhibitors dissolved in DMSO (Cdc7i: 15µM PHA-767491 Selleck Chemicals S2742, Cdk2i: 10µM NU-6140 Cayman Chemical Company 17271, APCi: 20µM Thermo Fisher Scientific I44001M, SCFi: 3µM MLN4924 Selleck Chemicals S7109) were added either at 18hpi (Early) or 48 hpi (Late). Proteasome inhibition was performed by treating cells with 0.5µM bortezomib (Selleck Chemicals S7109) for 6 hours prior to lysate collection.

## siRNA knockdowns

~80% confluent RPTE cells were transfected with 10 nM siRNA against MCM2-7 (MCM2: ThermoFisher s8586, MCM3: ThermoFisher s8591, MCM4: ThermoFisher s8592, MCM5: ThermoFisher s8596, MCM6: ThermoFisher s8599, MCM7: ThermoFisher s224035) or equimolar siNTC (Ambion, 4390843) using Lipofectamine RNAiMAX (ThermoFisher, 13778150). Briefly, Lipofectamine RNAiMAX and the siRNA were added to media in 12-well. RPTE cells were trypsinized and suspended to a concentration of $1.75 \times 10^5$ cells/ml and added to the siRNA containing media and incubated at 37°C/5% $CO_2$. The following day, the transfection media was replaced with fresh REGM and incubated at 37°C/5% $CO_2$ for 48 hours prior to infection.

## Western analyses

Proteins lysates were collected using E1A lysis buffer and quantified by Bradford assay (Thermo Fisher Scientific, #23236). Lysates were separated using 12% SDS-PAGE, transferred to polyvinylidene difluoride (Millipore Sigma #IPFL00010), and blocked using 5% milk with 0.1% Tween-20 in PBS. Protein targets were probed at 1:2000 in 5% milk/PBST (TAg: pAb416 as a gift from Dr. Mengxi Jiang, VP1 and VP2/3: a gift from Dr. Mengxi Jiang, MCM2: sc-373702 Santa Cruz Biotechnology RRID:AB_10917436, phospho-MCM2[S53]: Abcam ab109133 RRID:AB_10863901, MCM3: Santa Cruz Biotechnology sc-390480, MCM4: Cell Signaling Technologies #3228 RRID:AB_11178393, beta actin: Cell Signaling Technologies #4967 RRID: AB_330288, Cdk2: Abcam ab32147 RRID:AB_726775, phospho-Chk1[S318]: Cell Signaling Technologies 12302S RRID:AB_2783865, phospho-ATM[S1981]: AbCam ab81292 RRID: AB_1640207, Cdc6: Cell Signaling Technologies 3387T RRID:AB_2078525, Cdt1: Cell Signaling Technologies 8064S RRID:AB_10896851, Geminin: Abcam ab195047 RRID:AB_2832993, Tubulin: Cell Signaling 2146 RRID:AB_2210545, Erk1/2: Thermo Fisher Scientific ERK-7D8 RRID: AB_2533024, phospho-Erk1/2 (T202/Y204): Cell Signaling 9101S RRID:AB_331646) and visualized either using LiCOR secondary antibodies (LI-COR Biosciences, anti-Mouse IR 800 #926–32212 or anti-Rabbit IR680, #926–68073) on the LiCOR Odyssey Fc, or HRP-conjugated secondary antibodies (GE Healthcare anti-mouse NA931 RRID:AB_772210 or GE Healthcare anti-rabbit NA934 RRID:AB_772206) developed using Luminata Forte Western HRP substrate (Millipore WBLUF0100) and Midsci chemiluminescence film (BX810). Band quantification was performed from westerns using the LiCOR secondary antibodies with LiCOR Image Studio.

## Immunofluorescence microscopy

After removing the media, RPTE cells were fixed in 4% PFA (Electron Microscopy Sciences C993M23) in PBS for 15 minutes, washed in PBS 3X, and permeabilized for 5 minutes using 0.3% Triton-X100 (Millipore Sigma, T8532) in PBS. For experiments using 5-Ethynyl-2′-deoxyuridine (EdU, Click Chemistry Tools #1149–25), the permeabilized cells were treated with Alexa-Fluor 488 azide (Click Chemistry Tools #1275–5) in PBS with 100 mM CuSO$_4$ (Sigma Aldritch C1297), 500 mM sodium ascorbate (ACROS Organics 352681000), and rocked for 1 hour at room temperature in the dark followed by two PBS washes. Prior to antibody probing, cells were blocked using 5% FBS in PBS for 30 minutes at room temperature. Probing with primary antibodies (1:250 in 5% FBS in PBS) overnight at 4˚C (TAg: pAb416 as a gift from Dr. Mengxi Jiang, VP2/3: gift from Dr. Mengxi Jiang, MCM4: Cell Signaling Technologies #3228 RRID:AB_11178393, phospho-MCM2[S53]: Abcam ab109133 RRID:AB_10863901, phospho-H3[S10]: Cell Signaling #3458S RRID:AB_10694086, phospho-Chk1[S318]: Cell Signaling Technologies 12302S RRID:AB_2783865, phospho-ATM[S1981]: Abcam ab81292 RRID:AB_1640207, BrdU: Thermo Fisher Scientific B35133 RRID:AB_2536437). Secondary antibodies (1:250 in 5% FBS in PBS) for 1 hour at room temperature (anti-mouse conjugated to DyLight 594: Novus Biologicals NBP1-75957, anti-mouse conjugated to AF647: Thermo Fisher Scientific A21235 RRID:AB_2535804, anti-rabbit conjugated to DyLight 594: Novus Biologicals NBP1-76061). DNA was visualized using FxCycle-violet (Invitrogen F10347, 1:1000 in PBS) just prior to imaging. Images were taken using the Keyence BZ-X810 equipped with Chroma filters at 10x magnification. Microscopy quantification was performed using FIJI ImageJ and analyzed using R (version 4.1.2) in RStudio (Posit, version 2022.12.0). The ImageJ and R scripts used for single-cell analysis are available at https://github.com/SThompsonLab/MicroCyte.

## Lentivirus production and transduction

HEK293 cells were transfected using Lipofectamine 2000 (ThermoFisher Scientific 11668019) in opti-MEM (ThermoFisher Scientific 31985088) with pBOB-EF1-FastFUCCI-Puro (a gift from Kevin Brindle & Duncan Jodrell, Addgene plasmid # 86849), psPAX2 (a gift from Didier Trono, Addgene plasmid # 12260), and pMD2.G (a gift from Didier Trono, Addgene plasmid #) at a ratio of 4:3:1 for a total of 24 μg per 10cm plate. Transfected HEK293 cells were grown in DMEM supplemented with 10% FBS and 2.5mM L-glutamine and the resulting lentivirus was concentrated on a 30% sucrose cushion at 100,000 x *g* for 90 minutes. Transduction of RPTE cells was performed by adding lentivirus to RPTE cells in REGM overnight and titering the lentivirus by focus-forming assay in RPTE cells at 5 days post-transduction.

## Flow cytometry

Prior to collection, Edu label was added to cell growth media (1:1000) and cells were incubated for 2.5 hours at 37˚C. Cells were trypsinized, washed in 1X PBS, and fixed in 4% paraformaldehyde, 1XPBS. Cells were permeabilized using 0.3% Triton in wash buffer (1% FBS in 1x PBS) for 15 minutes, washed in 1X PBS, and resuspended in 200 μl of click-it fluorophore fixation reagent (20 μM Alexa Fluor 488 azide, 2 mM CuSO4, 10 mM Na-ascorbate) and incubated 1 hour in the dark. Mitosis was visualized with a AF647 conjugated phospho-H3[S10] antibody (Cell Signaling #3458S RRID:AB_10694086; 1:100 in wash buffer) and DNA was stained using FxCycle Violet dye (Invitrogen F10347, 1:1000 in wash buffer). Data collection was performed on a LSR II flow cytometer using the 405, 488, and 647 laser lines and the FACSDIVA software for a minimum of 50,000 events. Gating and subsequent analyses were performed using R (version 4.1.2) in RStudio (Posit, version 2022.12.0).

## Statistical analyses

All replicates refer to biological replicates. Statistical analysis was performed using R (version 4.1.2) in RStudio (Posit, version 2022.12.0). For two-sample comparisons, a Student's t-test was used with an alpha set at 0.5. For more than two-sample comparisons, an ANOVA was used and, if p was found to be less than 0.5, a Tukey post-hoc test was used with an adjusted alpha of 0.5.

## Supporting information

**S1 Fig. MCM knockdown decreased BKPyV production.**
(DOCX)

**S2 Fig. MCM inhibition early, but not late, decreases TAg expression.**
(DOCX)

**S3 Fig. FastFUCCI transduced RPTE cells do not impair BKPyV infection.**
(DOCX)

**S4 Fig. ATR and ATM activation primarily found in re-replicating cells.**
(DOCX)

**S5 Fig. BKPyV binding during infection does not activate MAPK pathway.**
(DOCX)

## Acknowledgments

We would like to thank Dr. Mengxi Jiang for her donation of cells and viral stocks and Dr. Michael Imperiale (University of Michigan) for the pAB416 antibody.

## Author Contributions

**Conceptualization:** Jason M. Needham.

**Data curation:** Jason M. Needham.

**Formal analysis:** Jason M. Needham.

**Funding acquisition:** Jason M. Needham, Sunnie R. Thompson.

**Investigation:** Jason M. Needham, Sarah E. Perritt.

**Methodology:** Jason M. Needham.

**Project administration:** Jason M. Needham, Sunnie R. Thompson.

**Software:** Jason M. Needham.

**Supervision:** Jason M. Needham, Sunnie R. Thompson.

**Validation:** Jason M. Needham.

**Visualization:** Jason M. Needham.

**Writing – original draft:** Jason M. Needham.

**Writing – review & editing:** Jason M. Needham, Sunnie R. Thompson.

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
