## [Decision Letter · Decision Letter 0]

26 Jul 2024

Dear Dr. Thompson,

Thank you very much for submitting your manuscript "Single-cell analysis reveals host S phase drives large T antigen expression during BK polyomavirus infection" for consideration at PLOS Pathogens. As with all papers reviewed by the journal, your manuscript was reviewed by members of the editorial board and by several independent reviewers. In light of the reviews (below this email), we would like to invite the resubmission of a significantly-revised version that takes into account the reviewers' comments.  Please pay particular attention to the request by one reviewer that you include a critical control, that being virus like particle without encapsidated viral genomes, in your key experiments to distinguish between consequences of an active infection versus a mitogenic response to noninfectious empty virus particles.  

We cannot make any decision about publication until we have seen the revised manuscript and your response to the reviewers' comments. Your revised manuscript is also likely to be sent to reviewers for further evaluation.

Sincerely,

Paul F. Lambert

Academic Editor

PLOS Pathogens

Alison McBride

Section Editor

PLOS Pathogens

Michael Malim

Editor-in-Chief

PLOS Pathogens

orcid.org/0000-0002-7699-2064

Reviewer's Responses to Questions

**Part I - Summary**

Reviewer #1: PPATHOGENS-D-24-01180

The manuscript by Needham et al describes studies on the initial replication of BK virus (BKPyV) after infection of primary human kidney cells (RPTE). The major conclusion is that the viral large T-antigen (LTag) stimulation of host cell replication follows an initial host cell S-phase, which is unrelated to viral protein expression. Then, once the viral proteins are expressed following this initial S-phase induction, viral replication becomes directed by LTag and is independent of host cell replication. The data is convincing with respect to the expression of LTag for re-replication. However, as observed in infection by other polyomaviruses (e.g., SV40, MuPyV) virion binding to cells alone initiates a mitogenic response, inducing multiple serum response genes (e.g, c-myc, c-fos, etc.). Thus, although LTag is responsible for subsequent cell cycle disruption the virus appears to rely upon virion attachment to initiate infection. In a sense therefore the conclusions here are correct, but the initial activating source of cell cycle induction, which is the virus itself, is a missing part of the experiments and discussion. The mock controls shown are insufficient to address this important point, and a necessary control would be BKV VLPs (no genomes) to demonstrate the independent mitogenic activity of virus attachment.

Reviewer #2: This interesting study uses single cell analysis to examine the effect of cell cycle on large T antigen expression during BK polyomavirus infection. The accepted model in the field is that TAg is expressed early to promote cell cycle transition into S phase. Here they suggest that TAg expression requires host DNA replication. They propose a new model in which TAg expression depends upon an initial host S phase, and that the viral genome is replicated primarily during a subsequent phase of host re-replication. The manuscript is well written, the data are rigorous and clearly presented, and the interpretations are thoughtful and mostly supported by the data. They use a powerful combination of elegant approaches. There are some caveats, but these are thoughtfully considered. The results are likely to stimulate new directions of research in this area.

Figure 1 examines where TAg is expressed in cell cycle by EdU labeling and single cell cycle plots combined with immunofluorescence. TAg accumulates in the re-replicating cells. Could it be that TAg made in newly infected cells is less stable that TAg made in S-phase and this explains the difference in intensity and low amount at early times? This is partly addressed by later experiments but differential stability is not fully discussed.

In Figure 2 they use inhibitors to block the initial S phase and compare to inhibition at late times. These data suggest BKPyV requires the initial host S phase for virus production. Do they have controls to demonstrate that the inhibitors have been effective? The pulse labeling experiment in Fig 2D is particularly elegant. Have they done anything to analyze viral gene expression at the transcription level (e.g. FISH) during these different cell cycle stages?

Figure 3 uses increasing MOIs to account for viral genome copy number. Did they check that higher MOIs correlates with the expected amount of viral genomes uncoated in the nucleus? Have they considered that there could be a repressor preventing TAg gene expression in the TAg-ve cells? Have they also considered labeling nascent proteins and immunoprecipitating for TAg to see whether alterations in protein synthesis levels could be a contributing factor?

Figure 4 looks at the requirements of origin licensing factors and Figure 5 examines firing of new origins. These are elegant but somewhat complex experiments. The interpretations are cautious and suggest that canonical mechanisms are used for origin licensing and firing.

Figure 6 and the Discussion present their new model which requires S phase for TAg expression. Here they discuss alternative splice variants and truncated Tags. They could speculate a little more about what activates TAg expression in S/G2, e.g. specific transcription factors, altered translational control, loss of a repressor etc.

**Part II – Major Issues: Key Experiments Required for Acceptance**

Reviewer #1: A necessary control would be BKV VLPs (no genomes) added to the cells in order to demonstrate the independent mitogenic activity of virus attachment.

Reviewer #2: Described above

**Part III – Minor Issues: Editorial and Data Presentation Modifications**

Reviewer #1: Other Comments:

1. For BKV “reactivation”, it would be helpful to clarify (at least comment on, since likely unknown) where the virus is being reactivated from, and how the virus may be maintained in cells before reactivation. Although briefly addressed in the final sentences of the conclusions (lines 626-633), this biology seems important to the possible in vivo relevance of these experiments.

2. “Robust levels” of LTag are not explained more than immunofluorescent measurements. Another more quantitative assay for LTag would be useful. At the least, estimate the level of detection.

3. The figures are extremely complex (5 x 9 panels each!)! Although the data are appreciated, the presentation might/should be clarified. Perhaps move some related panels to the appendix, or combine related panels into unified figures?

3. A stylist comment: “this” without a subject is often confusing. Please correct.

4. References to similar experiments with other polyomaviruses would be helpful.

Reviewer #2: Line 273 is missing a word?

PLOS authors have the option to publish the peer review history of their article (what does this mean?). If published, this will include your full peer review and any attached files.

Reviewer #1: No

Reviewer #2: No
---

## [Decision Letter · Decision Letter 1]

11 Oct 2024

Dear Dr. Thompson,

We are pleased to inform you that your manuscript 'Single-cell analysis reveals host S phase drives large T antigen expression during BK polyomavirus infection' has been provisionally accepted for publication in PLOS Pathogens.

Best regards,

Paul F. Lambert

Academic Editor

PLOS Pathogens

Alison McBride

Section Editor

PLOS Pathogens

Michael Malim

Editor-in-Chief

PLOS Pathogens

orcid.org/0000-0002-7699-2064

Reviewer Comments (if any, and for reference):

Reviewer's Responses to Questions

**Part I - Summary**

Reviewer #1: The induction of host cell DNA synthesis by polyomavirus infection is a very interesting and important topic. The findings in this paper concerning BKV seem to contrast with those reported for other polyomaviruses where initial attachment of the virus to the cell surface induces a mitogenic response. Here, prior "normal" cell DNA replication seems to be required before the viral T-ag induces a re-replication phase. The control used in the resubmission is ERK kinase activity, which supports this conclusion. In general this paper provides important details on the initiation of BKV infection.

Reviewer #2: The authors have carefully considered all reviewers’ comments in this revised manuscript. They have provided thoughtful responses and modified the manuscript appropriately. They should be commended for this rigorous and thorough manuscripts, which is complicated but well presented.

**Part II – Major Issues: Key Experiments Required for Acceptance**

Reviewer #1: Additional experiments have been described that support the conclusions. The figures remain daunting to navigate, but the results are detailed and convincing.

Reviewer #2: (No Response)

**Part III – Minor Issues: Editorial and Data Presentation Modifications**

Reviewer #1: none

Reviewer #2: (No Response)

PLOS authors have the option to publish the peer review history of their article (what does this mean?). If published, this will include your full peer review and any attached files.

Reviewer #1: No

Reviewer #2: No

---

## [Editor Report · Acceptance letter]

25 Oct 2024

Dear Dr. Thompson,

We are delighted to inform you that your manuscript, "Single-cell analysis reveals host S phase drives large T antigen expression during BK polyomavirus infection," has been formally accepted for publication in PLOS Pathogens.

Best regards,

Michael Malim

Editor-in-Chief

PLOS Pathogens

orcid.org/0000-0002-7699-2064